# Effective Treatment of Patients Experiencing Primary, Acute HIV Infection Decreases Exhausted/Activated CD4+ T Cells and CD8+ T Memory Stem Cells

**DOI:** 10.3390/cells11152307

**Published:** 2022-07-27

**Authors:** Domenico Lo Tartaro, Antonio Camiro-Zúñiga, Milena Nasi, Sara De Biasi, Marco A. Najera-Avila, Maria Del Rocio Jaramillo-Jante, Lara Gibellini, Marcello Pinti, Anita Neroni, Cristina Mussini, Luis E. Soto-Ramírez, Juan J. Calva, Francisco Belaunzarán-Zamudio, Brenda Crabtree-Ramirez, Christian Hernández-Leon, Juan L. Mosqueda-Gómez, Samuel Navarro-Álvarez, Santiago Perez-Patrigeon, Andrea Cossarizza

**Affiliations:** 1Department of Medical and Surgical Sciences for Children and Adults, University of Modena and Reggio Emilia, 41125 Modena, Italy; domenico.lotartaro@unimore.it (D.L.T.); debiasisara@yahoo.it (S.D.B.); lara.gibellini@unimore.it (L.G.); anitaneroni@hotmail.it (A.N.); 2Department of Biomedical, Metabolic and Neural Sciences, University of Modena and Reggio Emilia, 41125 Modena, Italy; 3Instituto Nacional de Ciencias Médicas y Nutrición Salvador Zubirán, Infectious Diseases, Mexico City 14080, Mexico; antoniocamiro@hotmail.com (A.C.-Z.); marco.najera06@gmail.com (M.A.N.-A.); rousse_j16@hotmail.com (M.D.R.J.-J.); lesoto@hotmail.com (L.E.S.-R.); juanjcalva@gmail.com (J.J.C.); belaunzaranzapabf@niaid.nih.gov (F.B.-Z.); brenda.crabtree@infecto.mx (B.C.-R.); santiago.perez@queensu.ca (S.P.-P.); 4Department of Surgery, Medicine, Dentistry and Morphological Sciences, University of Modena and Reggio Emilia, 41124 Modena, Italy; 5Department of Life Sciences, University of Modena and Reggio Emilia, 41125 Modena, Italy; marcello.pinti@unimore.it; 6Infectious Diseases Clinics, Azienda Ospedaliero-Universitaria Policlinico di Modena, 41124 Modena, Italy; cristina.mussini@unimore.it; 7Centro Ambulatorio para la Prevención y Atención del Sida e Infecciones de Transmisión Sexual (CAPASITS), Puebla 72410, Mexico; chrishleon@hotmail.com; 8Centro Ambulatorio para la Prevención y Atención del Sida e Infecciones de Transmisión Sexual (CAPASITS), Leon 37320, Mexico; luis_mosqueda@yahoo.com; 9Hospital General de Tijuana, Tijuana 22000, Mexico; dr.samuel.navarro@gmail.com; 10Division of Infectious Diseases, Queen’s University, Kingston, ON K7L 3N6, Canada; 11National Institute for Cardiovascular Research—INRC, 40126 Bologna, Italy

**Keywords:** HIV, highly active antiretroviral therapy, T-lymphocyte subsets, B-lymphocyte subsets, cellular senescence, memory stem cells

## Abstract

Several studies have identified main changes in T- and B-lymphocyte subsets during chronic HIV infection, but few data exist on how these subsets behave during the initial phase of HIV infection. We enrolled 22 HIV-infected patients during the acute stage of infection before the initiation of antiretroviral therapy (ART). Patients had blood samples drawn previous to ART initiation (T0), and at 2 (T1) and 12 (T2) months after ART initiation. We quantified cellular HIV-DNA content in sorted naïve and effector memory CD4 T cells and identified the main subsets of T- and B-lymphocytes using an 18-parameter flow cytometry panel. We identified correlations between the patients’ clinical and immunological data using PCA. Effective HIV treatment reduces integrated HIV DNA in effector memory T cells after 12 months (T2) of ART. The main changes in CD4+ T cells occurred at T2, with a reduction of activated memory, cytolytic and activated/exhausted stem cell memory T (T_SCM_) cells. Changes were present among CD8+ T cells since T1, with a reduction of several activated subsets, including activated/exhausted T_SCM_. At T2 a reduction of plasmablasts and exhausted B cells was also observed. A negative correlation was found between the total CD4+ T-cell count and IgM-negative plasmablasts. In patients initiating ART immediately following acute/early HIV infection, the fine analysis of T- and B-cell subsets has allowed us to identify and follow main modifications due to effective treatment, and to identify significant changes in CD4+ and CD8+ T memory stem cells.

## 1. Introduction

Primary HIV infection (PHI) is characterized by several viral and immunological phenomena that determine the course of the disease, including early dysfunctional adaptative immune responses and the establishment of a viral reservoir [1]. ART initiated during PHI in part helps to lessen these events and is associated with the lowering of the viral set point and reservoir, reduced immune activation and inflammation, mitigation of peripheral T-and B-cell dysfunction, and a more favorable reconstitution of immune cells [2,3,4,5]. Despite the well-described benefits of immediate ART, persistent immune activation remains a hallmark of chronic HIV infection even in patients treated during PHI [6]. This in turn leads to immune-senescence and contributes to the early onset of several co-morbidities, including cardiovascular, metabolic and neurodegenerative diseases [7,8,9]. The immunovirological events pertaining PHI have been largely described using non-human primate models [10,11], as conclusive prospective human studies have proven difficult to execute due to the innate obstacles of detecting human patients during the pre-infection period and during the earliest stages of primary infection [12]. However, scanty data describe the phenotype and function of B and T cells in different stages of HIV infection and ART interruption [13,14,15]. In particular, untreated individuals displayed high percentages of CD4+ cytotoxic T lymphocytes (CTL) expressing perforin, granzyme, CD38 and CD57. These cells emerge early during HIV infection, correlate with acute viral load, and are associated with early viral load set point [4]. Furthermore, patients with PHI are characterized by CD107a+ IFN-γ+ CD4+ T cells that are similar to HIV-specific CD8+ CTL with killing capability [16]. Several pieces of evidence indicate a close cooperation between HIV-reactive CD4+ and CD8+ T cells to control HIV infection. 

PD-1 expression on CD4+ and CD8+ T-cells is associated to immune dysfunction [17]. Recent data has shown that PD-1 on peripheral memory CD4+ T-cell subsets in HIV-infected subjects is preferentially expressed on the surface of persistently infected cells. [18].

Regarding B cell subset, HIV-viremic individuals display inadequacy of the anti-HIV antibody response and the presence in periphery of low responsive B cells (CD21^low^CD38^low^) subset expressing the inhibitory receptor Fc-receptor-like-4 (FCRL4) [19]. Moreover, after initiation of ART, B cell numbers increase, and their dynamics in response are more closely related to those of CD4+ T cells [20,21]. 

Fine analysis of the phenotypes and functionality of the adaptive immune cells involved in the containment of the virus are a key element for the better understanding of HIV infection [22]. Moreover, identifying the dynamics of lymphocyte subsets during the PHI stage will expand our understanding of the immune activation and response, even in the chronic phase of infection. For these reasons, we designed this study, aimed to identify the long-term changes in the main subsets of T and B lymphocytes in patients with PHI who initiated ART during PHI.

## 2. Materials and Methods

### 2.1. Patients

We enrolled patients with PHI (defined as a Fiebig stage of ≤5 or as a Fiebig stage of 6 with a negative fourth-generation enzyme-linked immunosorbent assay for HIV in the previous 3 months) [23]. Each patient started ART at diagnosis and was follow-up for at least 12 months. During this period, patient therapies were not modified. For the viro-immunological analyses, venous blood was collected at baseline (T0) and at 2 (T1) and 12 months (T2) following ART initiation. This study was conducted in agreement with ethical recommendations of the Declaration of Helsinki and was approved by the Ethics Committee. All participants gave written informed consent.

### 2.2. Blood Samples Collection, Processing, and Storage

Peripheral blood mononuclear cells (PBMCs) were isolated from 30 mL of venous blood by density gradient centrifugation and stored in liquid nitrogen using standardized protocols in Mexico [24]. Frozen PBMC were first stored for up to 18 months, sent to Italy in dry ice by overnight courier, and kept in liquid nitrogen until use. Then, cells were thawed and washed twice with RPMI 1640 supplemented with 10% fetal bovine serum and 1% each of l-glutamine, sodium pyruvate, nonessential amino acids, antibiotics, 0.1 M HEPES, 55 μM β-mercaptoethanol and 0.02 mg/ml DNAse [25]. Thawed cells were used for sorting or T and B immunophenotyping.

### 2.3. Sorting of Naïve and Effector Memory CD4+ T Cells

A minimum of 10 million PBMCs was stained with anti-CD4 FITC (R&D, Minneapolis, MN, USA), -CD45RA-PECy7, -CCR7 PE (both from Biolegend, San Diego, CA, USA) and LIVE/DEAD Fixable Red Dead (ThermoFisher Scientific, Waltham, MA, USA). Viable naïve and effector memory T cells were identified as CD4+CD45RA+CCR7+ and CD4+CD45RA-CCR7-, respectively, and sorted with a purity > 98% by using a Bio-Rad eS3 Sorter (Bio-Rad, Hercules, CA, USA) (Appendix A) [26].

### 2.4. DNA Extraction and Quantification of HIV-DNA

We extracted total DNA from naïve and effector memory T-cell subsets using the QIAmp DNA Minikit (Qiagen, Hilden, Germany). Cellular HIV-DNA was measured by the Bio-Rad QX200 droplet digital (dd) PCR system (Bio-Rad, Hercules, CA, USA) in 14 patients. Reagents were purchased from Bio-Rad: ddPCR Supermix for Probes and the custom assay, previously described, spanning the long terminal repeat (LTR) region and the reference gene EIF2C1 (UniqueAssayID: dHsaCP2500349) [26]. Primers were LTRdir 5′-GCCTCAATAAAGCTTGCCTTGA-3′, LTRrev 5′-GGGCGCCACTGCTAGAGA-3′, LTRprobe 5′-FAM-CCAGAGTCACACAACAGACGGGCACA–BLACK HOLE-3′ [26]. According to manufacturer’s instructions, seven μL of purified genomic DNA were added to a final volume of 12 μL mixture containing 2X ddPCR Supermix for Probes, 1 μL of the ddPCR assay for the reference gene EIF2C1 and 1 μL of the custom LTR primers.

### 2.5. Immunophenotyping of T Lymphocytes Using Flow Cytometry

Based on sample availability, we could interrogate 14 patients at different time points using a minimum of 1 × 10^6^ of PBMCs per patient per stain. PBMCs were thawed as described above. All cells were washed with PBS and stained with PromoFluor-840 (Promokine, PromoCell, Heidelberg, Germany) in PBS for 20 min at room temperature (RT). Next, cells were washed with FACS buffer (PBS added with 2% FBS) and stained for 20 min at room temperature (RT) in staining buffer (50% of Brilliant Stain Buffer and 50% of FACS buffer) with the Duraclone IM T-cell panel (Beckman Coulter, Brea, CA, USA) including CD3, CD4, CD8, CD27, CD28, CD45, CD45RA, CD57, CD197 (CCR7), and CD279 (PD-1), which was added with another five fluorescent mAbs HLA-DR-BUV661 (Becton Dickinson, catalog no. 565073, clone G46-6), CD127-BV650 (Biolegend, catalog no. 351326, clone A019D5), CD25-BV785 (Biolegend, catalog no. 302638, clone BC96), CD95-BUV395 (Becton Dickinson, catalog no. 740306, clone DX2), and CD38-BUV496 (Becton Dickinson, catalog no. 612946, clone HIT2). Samples were acquired on a six-laser CytoFLEX LX flow cytometer (Beckman Coulter). All mAbs added to DuraClone IM T cells were previously titrated on human PBMCs and used at the concentration giving the best signal-to-noise ratio. CytoFLEX Daily QC and CytoFLEX Daily IR QC (Beckman Coulter, catalog no. B53230 and C06147) were used to track and adjust gain over time. CompBeads (Becton Dickinson, catalog no. 552843) were used for compensation. At least 5 × 10^5^ viable PBMCs were acquired per sample.

### 2.6. Immunophenotyping of B Lymphocytes Using Flow Cytometry

To evaluate the effect of ART on B-cell phenotypic profiles, we evaluated their subpopulation frequency at T0 and T2. The procedures were the same used for the immunophenotyping of T lymphocytes, and the stain was performed with Duraclone IM B cell panel (Beckman Coulter) added with the cell viability marker PromoFluor-840 (PromoCell).

### 2.7. Statistical Analysis

#### 2.7.1. Analysis of Clinical Data

Changes over time of continuous variables (CD4+ and CD8 T+ cell count, CD4:CD8 ratio, plasma viral load and HIV-DNA) were analyzed by Wilcoxon matched-pairs signed-rank test or Friedman test using PRISM 9 (GraphPad Software, San Diego, CA, USA).

#### 2.7.2. High-Dimensional Data Analysis of Flow Cytometry Data

Flow Cytometry Standard (FCS) 3.0 files were imported into FlowJo software version 10 (Becton Dickinson), compensated, and analyzed by standard gating to eliminate aggregates, doublets, and dead cells. Then, data from 5000 CD3+CD4+ T cells, 5000 CD3+CD8+ T and 2500 CD19+ B cells per sample were exported for further analysis in R using Bioconductor libraries CATALYST (version 1.12.2) and diffcyt (version 1.8.8) [27,28]. All data obtained by flow cytometry were transformed in R using hyperbolic arcsine (arcsinh x/cofactor) applying manually defined cofactors. Clustering and dimensional reduction were performed using FlowSOM (version 3.12) and UMAP (version 0.2.0) algorithms, respectively. B cells, CD3+CD4+ T cells, and CD3+CD8+ T cells were analyzed separately. A quality control was performed for B cells, CD3+CD4+ T cells, and CD3+CD8+ T cells to check inter-file signal drift to avoid shifts in clustering step between samples (technical and rather than biological variance), as reported in Appendix A. Differential cell populations abundances analysis was performed by using patient pairing generalized linear mixed model (GLMM) implemented within diffcyt package applying the FDR cutoff ≤ 0.05, each p-value was reported in the respective figure. All statistically significant results obtained using the unsupervised method were also tested by manual gating and confirmed by applying the Mann–Whitney *t*-test or Kruskal-Wallis. The absolute number of CD4+ and CD8+ T cells was obtained by multiplying the cluster percentages and CD4 or CD8 cell count.

#### 2.7.3. Principal Component Analysis and Correlation Plots

The table containing the percentages of all clusters of CD3+CD4+, CD3+CD8+ T and CD19+ B cells (obtained by the unsupervised analysis) was employed for the principal component analysis (PCA) performed using Bioconductor libraries pca3d (version 0.10.2) and factoextra (version 1.0.7). The T0 and T2 time-points were used for this purpose. For each variable, the percentage contributing to PC1 and PC2 was also calculated. 

Correlation analysis was performed using T- and B-cell percentages and seven hematological and clinical parameters, i.e., age, Fiebig scale, viral load, body mass index (BMI), CD4+ and CD8+ T cell count. Pairwise correlations between variables were calculated and visualized as a correlogram using R packages stats (version 3.6.2) and corrplot (version 0.90). Spearman’s rank correlation coefficient (ρ) was indicated by color scale; significance was indicated by asterisks (* *p* < 0.05; ** *p* < 0.005; *** *p* < 0.0005). 

All raw data supporting the conclusions of this article are available upon request.

## 3. Results

### 3.1. Enrolled Patients and Clinical Data

We enrolled 22 patients (21 male) with acute/early HIV infection. The mean age was 35.1 ± 9.9 years. Using the Fiebig staging system, 4 patients were classified stage II, 1 stage III, 12 stage IV, 4 stage V and 1 stage VI. ART therapies included TDF/FTC/EFV (n = 14), TDF/FTC/RAL (n = 4), ABC/3TC/RAL (n = 2), ABC/3TC/DTG combination (n = 1) and DRV/r/TDF/FTC (n = 1). The main clinical manifestations in patients at enrollment were fever (85.7%), malaise (58.9%), fatigue (48.2%), pharyngitis (44.6%), lymphadenopathy (42.9%), arthralgia (41.1%), rash (37.5%), vomiting (35.7%), diarrhea (26.8%), weight loss (25.0%), night sweats (25.0%), and oral ulcers (10.7%). As expected, after twelve months of therapy, CD4 T-cell count (Figure 1A) and CD4:CD8 ratio (Figure 1B) increased significantly, while CD8 T-cell count (Figure 1A) tended to decrease, despite the difference T0–T2 being at the threshold of statistical significance (*p* = 0.051). Antiviral therapy was highly successful, as all patients but one attained undetectable (i.e., below 50 copies HIV RNA/mL) viral load (Figure 1C). Indeed, as reported in the magnification of Figure 1C, one patient had a viral blip of 292 copies/mL, that was not present in the following period of observation.

### 3.2. Total HIV-DNA Decrease after ART in T_EM_ Cells

The HIV-DNA content expressed as LTR/1000 cells was measured at T0 and T2 (Figure 1D). HIV-DNA was higher in T_EM_ than in naïve T cells at T0 and decreased significantly at T2, reaching a similar level of content to naïve T cells. Biological samples at T1 at our disposal were not sufficient to perform the sorting of CD4+ T-cell subsets and the quantification of HIV-DNA.

### 3.3. Immunophenotyping of T Lymphocytes 

The gating strategies for the identification of CD4+ and CD8+ T-cell subsets are reported in Appendix A.

#### 3.3.1. T_CM_ and T_SCM_ Cells Reduction within CD4 T Cell after 12 Months of ART 

The 2D visualization of CD4+ T cells performed by Uniform Manifold Approximation and Projection (UMAP) is reported in Figure 2A (showing graphs obtained by UMAP and FlowSOM). After unsupervised clustering, 12 main subsets were identified within CD4+ T cells, and their percentages are reported in the heatmap of Figure 2B.

Subsets of CD4+ T cells have been defined as follows: naïve CD38− [CD45RA+CCR7+CD27+CD28+CD38−], naïve CD38+ [CD45RA+CCR7+CD27+CD28+CD38+], T stem cell memory (T_SCM_) expressing CD38+DR+PD1+ [CD45RA+CCR7+CD27+CD28+CD95+CD38+HLA-DR+PD1+], central memory (T_CM_) CD38−PD1− [CD45RA−CCR7+CD27+CD28+CD38−PD1−], T_CM_ CD38+ PD1− [CD45RA−CCR7+CD27+CD28+CD38+PD1−], T_CM_ CD38+ PD1+ [CD45RA−CCR7+CD27+CD28+CD38+PD1+], transitional memory (T_TM_) CD38−DR− [CD45RA−CCR7−CD28+CD38−HLA-DR−], T_TM_ CD38+DR+ [CD45RA−CCR7−CD28+CD38+HLA-DR+], effector memory (T_EM_) CD38+DR+ [CD45RA−CCR7−CD28−CD38+HLA-DR+], T_EM_ CD38−DR−CD57+ [CD45RA−CCR7−CD28−CD38−HLA-DR−CD57+], T_EM_ CD38+DR+CD57+ [CD45RA−CCR7−CD28−CD38+HLA-DR+CD57+] and effector memory re-expressing CD45RA (T_EMRA_) [CD45RA+CCR7−CD28−CD57+] (for more details, see Appendix A). Using the generalized linear mixed models (GLMM; under diffcyt), we identified the differences in the distribution of clusters over time (Figure 2C): at T1 the percentage and the absolute number of the above-mentioned subsets remained similar, and at T2 only T_EM_ CD38+DR+ cells decreased significantly (Figure 2C and Appendix A). Between T1 and T2 the percentage and absolute number of T_SCM_, T_CM_ CD38+PD1−, T_EM_ CD38+DR+ cells, T_EM_ CD38+DR+CD57+ cells decreased, while that of T_TM_ CD38−DR− cells increased. We then focused our attention on a subset of effector memory (CCR7−CD45RA−CD27−CD28−CD57+) CD4+ T cells that has cytotoxic characteristics and displays markers of activation (i.e., that was strongly positive to HLA-DR and CD38), and, by using manual gating, we found that their percentage significantly decreases during therapy (Figure 2D and Appendix A). Interestingly, their counterpart, i.e., the same subset that was not activated (HLA-DR− and CD38−), at time 0 showed a negative correlation with plasma viral load (Appendix A) but remained stable during treatment (Figure 2D, Appendix A).

#### 3.3.2. ART Is Associated with a Huge Reduction of CD8+ T Cell Activation and T_SCM_

As reported in Figure 3A, applying the high dimensional data approach to CD8+ T cells, we identified 14 main subsets defined as naïve CD38− [CD45RA+CCR7+CD27+CD28+CD38−], naïve CD38+ [CD45RA+CCR7+CD27+CD28+CD38+], T_SCM_ expressing CD38+DR+PD1+ [CD45RA+CCR7+CD27+CD28+CD95+CD38+HLA-DR+PD1+], T_CM_ CD38+DR+ [CD45RA−CCR7+CD27+CD28+CD38+HLA-DR+], T_TM_ DR+ [CD45RA−CCR7−CD28+HLA-DR+], T_TM_ DR− [CD45RA−CCR7−CD28+HLA-DR−], T_EM_ CD38+DR+CD57− [CD45RA−CCR7−CD28−CD38+HLA-DR+CD57−], T_EM_ CD38+DR+CD57+ [CD45RA−CCR7−CD28−CD38+HLA-DR+CD57+], T_EM_ CD57+PD1− [CD45RA−CCR7−CD28−CD57+PD1−], T_EM_ CD57+PD1+ [CD45RA−CCR7−CD28−CD57+PD1+], T_EM_ CD57−PD1− [CD45RA−CCR7−CD28−CD57−PD1−], T_EMRA_ CD38−CD57+ [CD45RA+CCR7−CD28− CD38−CD57+], T_EMRA_ CD38−CD57− [CD45RA+CCR7−CD28−CD38−CD57−] and T_EMRA_ CD38+CD57+ [CD45RA+CCR7−CD28− CD38+CD57+] (for more details see Appendix A). These subsets clustered in three major islands, according to the expression of their surface markers, comprising naïve-T_SCM_ cells (on the left), activated cells expressing CD38 and/or HLA-DR (lower-right corner) and exhausted-like cells expressing CD57 and/or PD1 (upper-left corner). Figure 3C shows the changes in the distribution of these subsets in terms of percentage and absolute count during follow-up (see also Appendix A). At T1 there was an increase in the percentage but not in the absolute count of naïve CD38−, T_TM_ HLA-DR+, T_EM_ CD57+PD1+ and T_EM_ CD57−PD1− and T_EMRA_ CD38−CD57− subsets and a decrease of T_EM_ CD38+HLA−DR+CD57− in both percentage and absolute count (Figure 3C). Furthermore, the frequency of T_TM_ HLA-DR− cells was increased while T_SCM_, T_EM_ CD38+HLA-DR+CD57−, and T_EM_ CD38+HLA-DR+CD57+ was reduced among T2 CD8 T cells from AHI patients compared to T0 both in terms of percentage and absolute number. These data suggest a durable reduction of CD8 T-cell activation (Figure 3C). Finally, no differences were found between T1 and T2 except for T_CM_ CD38+HLA-DR+, which decreased significantly. Comparable results to unsupervised analysis were also obtained by using classical manual gating analysis as reported in Appendix A.

### 3.4. Immunophenotyping of B Lymphocytes 

The gating strategy for the identification of B cell subsets is reported in Appendix A.

#### Plasmablasts and Exhausted B Cell Reduction Occurs following ART 

Applying a high dimensional data approach, we identified 11 main clusters of B cells defined as naïve [CD27−CD38−CD21+CD24+IgD+IgM+], naïve IgD− [CD27−CD38−CD21+CD24+IgD−IgM+], memory unswitched [CD27+CD38−CD24+IgD+IgM+], memory switched [CD27+CD38−CD24+IgD−IgM−], memory IgM only [CD27+CD38−CD24+IgD−IgM+], plasmablasts IgM− [CD27+CD38+IgM−], plasmablasts IgM+ [CD27+CD38+IgM+], transitional [CD27−CD38+CD21+CD24+IgD+IgM+], exhausted IgD+IgM+ [CD27−CD38−CD21−CD24−IgD+IgM+], exhausted IgD−IgM+ [CD27−CD38−CD21−CD24−IgD−IgM+] and surface-negative B cells [CD27−CD38−CD21−CD24−IgD−IgM−] (Figure 4A,B). The so called “surface-negative” B cells are CD19+ cells that do not express any of the other markers used (for more details see Appendix A). As reported by Figure 4C, after 1 year of therapy, an increase of memory-switched B cells and a decrease of plasmablasts IgM+ and exhausted IgD−IgM+ was observed. We also analyzed B lymphocytes by manual gating and found results comparable to those obtained by unsupervised analysis (see Appendix A).

### 3.5. PCA of CD8 T Cells Identifies Critical Biomarkers Associated with Clinical Outcome

To explore relationships between dependent variables among immunological and clinical parameters, we applied PCA. Due to the lack of T1 time-point in the analysis of B cells, only T0 and T2 were considered. As shown in Figure 5A (source data are reported in Appendix A), the score plot of the first two principal components describes 19.3% (PC1) and 13.5% (PC2) of the total variance in the data. The leading principal components separate patients at T0 and T2 along PC1. Figure 5B shows the variables responsible for clustering: CD8+ T_EM_ CD38+HLA-DR+CD57− and memory unswitched B cells are positively correlated variables (at the same side of the plot), that are negatively correlated to memory-switched B cells and CD4+ T_EM_ CD38−DR−CD57+. 

Clinical and phenotypic data were explored by pairwise Spearman rank correlation test, and the results were visualized using a correlogram (Figure 5C,D).

As shown in Figure 5C, at diagnosis we found a negative correlation between HIV viral load and the proportion of a population of cytotoxic CD4+ T cells that had the features of effector memory cells and were CD38−HLA-DR−CD57+ (R = −0.61; *p* = 0.017). Moreover, as reported by Figure 5D, after 1 year of therapy we also observed a negative correlation between the CD4 T-cell count and the plasmablasts IgM− (R = −0.65; *p* = 0.017).

## 4. Discussion

In this study we analyzed the main subsets of T and B-lymphocytes in patients with PHI who initiated ART in this phase using 18-parameter flow cytometry and droplet digital PCR. Afterwards, we conducted a high-dimensional analysis of flow cytometry data to identify clusters of T- and B-cell populations, paying particular attention to cytotoxic CD4+ T cells, and we performed principal component analysis to determine correlations amongst the study variables. 

It is known that early ART initiation determines an increase of CD4+ T cell count and of the CD4:CD8 ratio, a decrease of CD8+ T-cell count and undetectable plasma viremia [29,30]. ART also induces several persistent changes in both pro-viral DNA and T- and B-lymphocyte subsets [31]. Accordingly, in our study the viral reservoir, measured as cellular HIV-DNA, dramatically decreased after 12 months of treatment, reaching in memory T cells a level of content like that in naïve T lymphocytes. A recent study found that pre-ART HIV-DNA, measured in CD4+ T cells, was the best predictor of HIV-DNA levels in patients with acute HIV infection after 12 months of treatment [32]. The level of pre-ART HIV-DNA in turn was associated with three main CD8+ T-cell subsets: memory, activated and Tim-3+ memory T cells, but the authors did not find any association among these parameters after 1 year of ART [32]. In our study, no correlations between HIV-DNA content and other clinical or immunological parameters, including pre- and post-ART, were found. This is probably due, at least in part, to the limited sample size of our study and the different subsets considered for HIV-DNA quantification. 

The depletion of CD4+ T cells, the key players of the adaptive immune response, is the hallmark of HIV infection and disease progression, while CD8+ T cells play a central role in the control of HIV replication. The fine analysis of T-cell subsets in patients with acute/early infection before and after therapy could shed light on the dynamics of the subsets redistribution and help to identify those with a greater significance in HIV control. We found that the main changes in CD4+ T-cell subsets occurred between 2 and 12 months after therapy initiation; as expected, all the activated memory CD4 T-cell subsets, including T_SCM_ CD38+DR+PD1+ and activated, effector memory cytotoxic CD4+ T cells, showed a decrease, independently from the expression of CD57 or PD1 surface marker. Recently, a distinct CD4+ T-cell subset capable of exerting a cytolytic activity has emerged as an important player in the control of HIV infection [17,33]. These cells have been found positive for perforin, granzyme and CD57. We found two clusters of cytotoxic CD4+ T cells expressing CD57, discerned based upon CD38 and HLA-DR expression. Interestingly, non-activated, effector memory CD38−DR−CD57+ cytolytic CD4+ T cells inversely correlates with HIV viral load at diagnosis, suggesting a potential cooperative role for these cells in controlling the first phases of HIV infection. The fact that the same functional subset expressing markers of activation decreases with therapy allows us to hypothesize a role of such cells in maintaining lymphocyte activation or even inflammation. 

It should be noted that, besides a tendency of most subsets to decrease or to remain stable (in terms of percentage and absolute count), CD4+ T-cell count increased; this is probably because naïve CD38− and T_CM_ CD38−PD1− cells, detected as the two larger subsets (64.9% of cells in total), showed a trend towards an increase, although not statistically significant. Such variations in CD4+ memory T cells may result from a progressive contraction of the adaptive immune response after the acute/early phase, in part. Moreover, we reported first that the therapy causes a decline in T_SCM_ CD38+DR+PD1+ (in terms of percentage and absolute count). This could be of interest, since it was recently reported that, during the acute phase, CD4+ T_SCM_ cell expansion is associated with both productively infected CD4+ T cells and the progression of the disease [34]. T_SCM_ cells are a preferred target of the virus that exploits the self-renewal ability of these cells to establish a persistent reservoir [35,36].

At the same time, we observed that the variations of CD8+ T cell subsets occurred earlier than those of CD4+ T cells in several subsets, since most of the differences were observed between T0 and T1, and these changes were maintained at T2. We observed an increase of naïve CD38−, T_TM_, non-activated T_EM_ and T_EMRA_ cells not expressing CD38 or CD57 in term of percentages but not in the absolute count. Interestingly, the low proportion of CD8+ T cells expressing CD57 but not CD28 predicts increased mortality in early ART-treated subjects [37]. On the contrary, the percentage and the absolute count of each memory subset expressing the CD38 surface antigen decreased at T1 or T2 compared to T0. The lack of significant differences between T1 and T2 could indicate that the therapy induces a quick and lasting reduction of CD8 T-cell activation. As for CD4+ T cells, the therapy caused a decrease in the absolute count of CD8+ T_SCM_ CD38+DR+PD1+, an exhausted subset characterized by the incapacity to generate functional cells [38]. To our knowledge, no data are available on the changes of this subset in acute/early HIV infection. However, it is known that CD8+ T_SCM_ were decreased in patients with chronic, untreated HV infection and restored by ART [39]. These changes in patients with acute/early infection mirror those observed in late presenters (patients experiencing a late diagnosis of HIV infection) that initiate ART. After 6 months of ART, late presenters showed an increase of CD4+ T cells and CD4/CD8 T cell ratio, in CD8+ naïve T cell and a significant reduction of CD4+ and CD8+ effector memory and activated T cells [40].

Several studies have reported alterations in B cell response during HIV infection, with a redistribution of B cell subsets characterized by the accumulation of differentiated B lymphocytes [40] and excessive B cell activation [41]. Effective antiretroviral therapy can partially or completely normalize B-cell defects, causing an increase in naïve B cells [31] and a decrease in activated B lymphocytes [42]. Concerning our study, the most notable change in B-cell subsets was within the memory switched cells. These cells, which are markers of a normal B-cell activation, increase 12 months after therapy initiation, while plasmablasts and exhausted B cells expressing IgM decrease. This result probably indicates that efficient ART leads to a normal B-cell response. Indeed, ART may also reduce the frequency of plasmablasts, despite the change being insufficient for their normalization when compared to healthy individuals [43]. It should also be noted that, among the main clusters identified by high-dimensional data analysis, the surface B-cell cluster could represent an autoreactive B-cell subpopulation with the capability to differentiate in Ig-secreting autoreactive plasma cells. It has been reported that these autoreactive cells express high levels of CD11c and T-bet but low expression of CD21, CD27, CD38, CD24, IgM and IgG [44]. We are aware that, in our panel, the absence of CD11c and T-bet did not allow us to identify with certainty this subpopulation, but these cells represent the only cluster of B cells simultaneously negative for the aforementioned markers. 

Finally, we found that plasmablasts IgM- were inversely correlated to CD4+ T-cell count following 12 months of therapy. Plasmablasts are the precursors of plasma cells and secrete IgM, but they can also produce low-affinity isotype-switched Abs. Following activation, plasmablasts usually undergo apoptosis within a few days, but their lifespan can be prolonged by the persistence of a pathogen [45]. We did not observe a decrease of IgM-negative plasmablasts after 1 year of therapy; however, they seem to be related to the increase of CD4+ T-cell count, probably indicating the contraction of an effective immune response. Further studies need to be performed to evaluate if the percentage of plasmablasts IgM− is a marker that anticipates or predicts CD4 T-cells’ recovery.

In conclusion, the fine analysis of T- and B-cell subsets with new and sophisticated methods of analysis has allowed us to identify the redistribution of the main subsets in patients initiating treatment immediately following acute/early HIV infection diagnosis and that of exhausted/activated CD4 and CD8 T memory stem cells.

## Figures and Tables

**Figure 1 cells-11-02307-f001:**
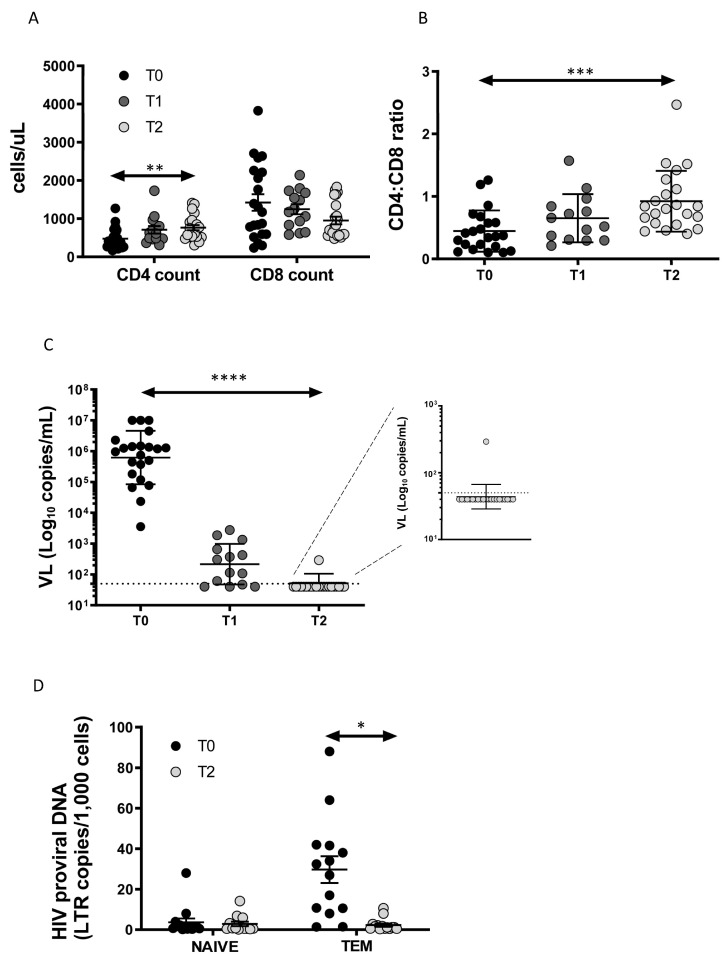
Immunovirological data: (**A**) CD4+ and CD8+ T-cell count at diagnosis (T0), at 2 (T1) and 12 months (T2) after the start of therapy; (**B**) CD4:CD8 ratio increases significantly after 12 months of therapy; (**C**) effective HIV treatment results in a decrease in viral load; (**D**) HIV-DNA content decreases in T_EM_ cells. *p*-values are indicated by white asterisks (∗ *p* < 0.05, ∗∗ *p* < 0.01, ∗∗∗ *p* < 0.001, ∗∗∗∗ *p* < 0.0001).

**Figure 2 cells-11-02307-f002:**
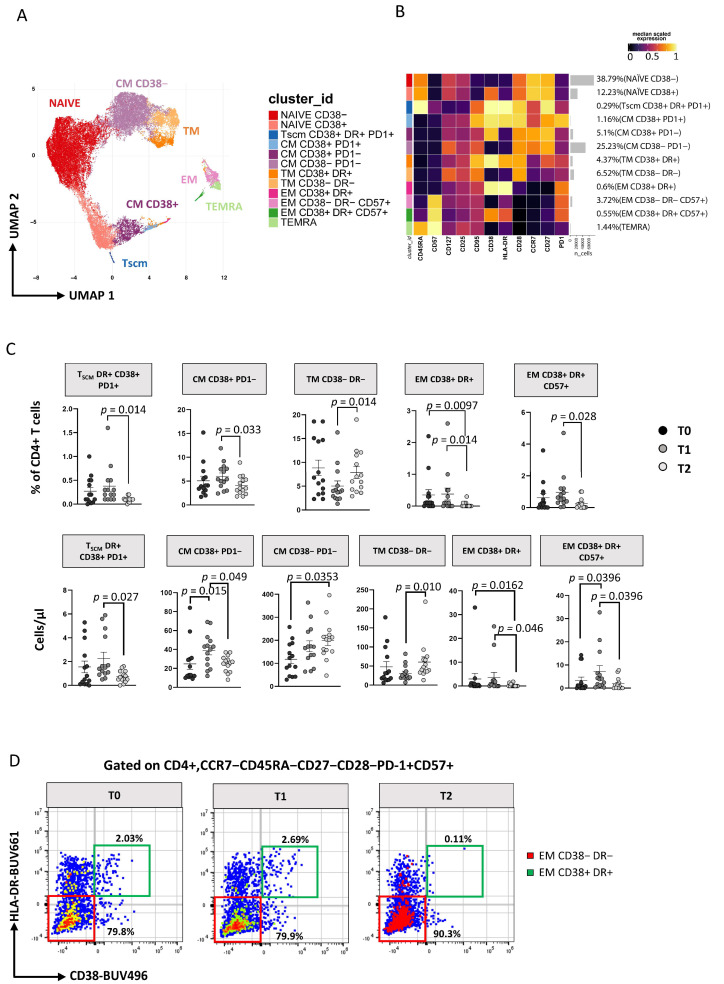
ART induces time-dependent changes in CD4+ T-cell subsets. (**A**) Two-dimension visualization of the CD4+ T-cell landscape by UMAP; (**B**) heatmap shows a median-scaled expression of markers across the CD4+ T-cell clusters. Bar plots along the rows and values in brackets on the right indicate the relative sizes of clusters. Twelve clusters were identified by FlowSOM using CD45RA, CD57, CD127, CD25, CD95, CD38, HLA-DR, CD28, CD27 and PD1 markers; (**C**) dot plots show the relative cells percentage (upper) and the absolute number (bottom) of the statistically significant cell clusters at T0 (black circles; n = 14), T1 (grey circles; n = 14) and T2 (light-grey circles; n = 14). The central bar represents the mean ± SEM. The adjusted *p*-values were obtained by patient pairing generalized linear mixed model (GLMM) that compares cluster percentages or absolute number at T0, T1 and T2, and are reported in the figure (threshold: FDR ≤ 0.05); (**D**) Representative flow cytometry dot plots showing manual gating analysis of EM CD38−DR−CD57+, and CD38−DR−CD57+ CD4+ T cells at T0, T1, and T2. Numbers in the dot plots indicate the percentage of cells identified by the gates.

**Figure 3 cells-11-02307-f003:**
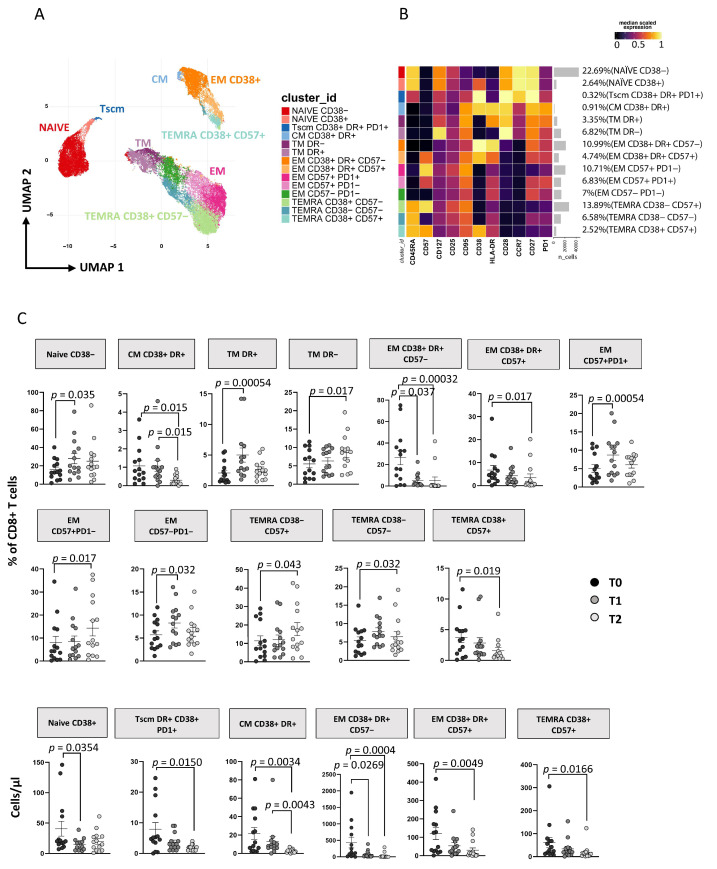
ART induces time-dependent changes in CD8+ T cell subsets. (**A**) Two-dimension visualization of the CD4+ T-cell landscape by UMAP; (**B**) heatmap shows a median-scaled expression of markers across the CD4+ T-cell clusters. Bar plot along the rows and values in brackets on the right indicate the relative sizes of clusters. Twelve clusters were identified by FlowSOM, using CD45RA, CD57, CD127, CD25, CD95, CD38, HLA-DR, CD28, CD27 and PD1 markers; (**C**) dot plots show the relative cells percentage (upper) and absolute number (bottom) of the statistically significant cell clusters at T0 (black circles; n = 14), T1 (grey circles; n = 14) and T2 (light-grey circles; n = 14). The central bar represents the mean ± SEM. The adjusted *p*-values were obtained by patient pairing generalized linear mixed model (GLMM) that compares cluster percentages or absolute numbers at T0, T1 and T2, and are reported in the figure (threshold: FDR ≤ 0.05).

**Figure 4 cells-11-02307-f004:**
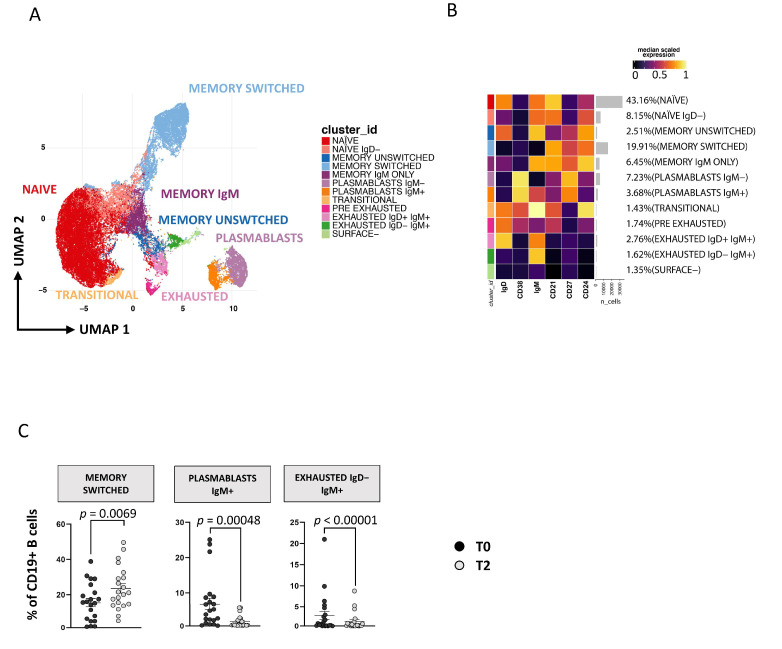
Time-dependent changes in CD19+ B-cell subsets. (**A**) Two-dimensional visualization of the CD19+ B-cell landscape by UMAP. (**B**) Heatmap shows a median-scaled expression of markers across the clusters. Bar plot along the rows and values in brackets on the right indicate the relative sizes of clusters. Twelve clusters were identified by FlowSOM using IgD, CD38, IgM, CD21, CD27 and CD24 markers. (**C**) Dot plots show the relative cells percentage of the 12 clusters from T0 (black circles; n = 21) and T2 (light-grey circles; n = 21) conditions. The central bar represents the mean ± SEM. The statistical relevant adjusted *p*-values obtained by patient pairing GLMM statistical test comparing T0 and T2 cluster percentages are reported in the figure (threshold: FDR ≤ 0.05).

**Figure 5 cells-11-02307-f005:**
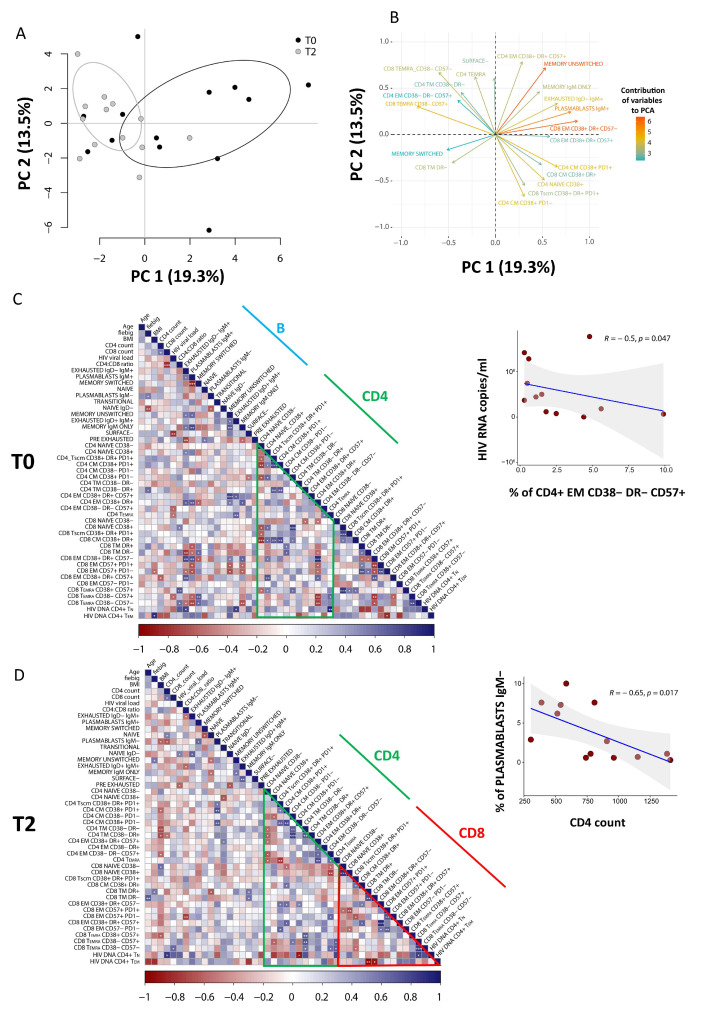
Contribution of different variables and correlations between clinical and immunological data. (**A**) Principal component analysis (PCA) of patients performed using T and B frequencies at T0 (n = 13) and T2 (n = 13), as obtained from high-dimensional data analysis. (**B**) Weight of the first 20 most important variables that contribute to PCA. Positively correlated variables point to the same side of the plot, whilst negatively correlated variables point to opposite sides of the graph. (**C**,**D**) Correlograms of patients before starting (T0) and after 12 months of ART (T2). Percentage of B and T clusters, intracellular HIV-DNA content and clinical parameters were used to perform the analysis. Spearman rank correlation values (r) are shown from red (−1.0) to blue (1.0); not-ordered r values are indicated by the color intensity of the square. Blank square indicates lack of signal; p-values are indicated by white asterisks (∗ *p* < 0.05, ∗∗ *p* < 0.01, ∗∗∗ *p* < 0.001). The Figure also shows the correlation of EM CD38−DR−CD57+ (indicated as percentage of total CD4+ T cells) with plasmatic viral load (copies/ml) at diagnosis (T0) and the correlation of IgM- plasmablasts (indicated as percentage of total CD19+ B cells) with the absolute count of CD4+ T cells after 12 months of therapy.

## Data Availability

All raw data supporting the conclusions of this article are available upon request.

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
