# Peer review of "Effective Treatment of Patients Experiencing Primary, Acute HIV Infection Decreases Exhausted/Activated CD4+ T Cells and CD8+ T Memory Stem Cells"

_cells, 2022, doi:10.3390/cells11152307_

Round 1

Reviewer 1 Report

In this manuscript, Lo Tartaro and colleagues analyze T and B cell population 22 HIV patients during the acute phases of infection and follow them 2 and 12 months after initiation of therapy.

Determining how T and B cell populations changes after therapy is an important topic for the field. Moreover, the cohort, even though small, is very peculiar as not many studies are performed in patients during acute HIV infection.

The authors show how the most significant changes in CD4 T cell subsets composition occur 12 months after therapy while changes in CD8 are already present 2 months after treatment. Specifically, they observed a reduction of activated memory and exhausted activated TSCM in the CD4 T cell compartment, while the reduction in activated CD8 was detectable at the early time point. Significant changes were also observed for B cells 12 months after initiation of ART.

Of note, a correlation between viral load and a specific subset of cytotoxic CD4 effector memory cells as well percentage of IgM plasmablast and CD4 T cell count was observed.

This study is well done and the data support the conclusions.

However, the manuscript is hard to follow due to the lack of information in the introduction. The authors should improve it. In paragraph 3.3.1_2_3, Lo Tartaro and colleagues describe all the subsets they found through clusterization, what are those? Why did they choose those PD-1, CD38 and CD57. A little bit of background would help the reader to understand the importance of their findings. 

Why did the authors completely disregard monocytes? Monocytes contribute to both HIV persistence and pathogenesis.

With the exception of TSCM, the authors analyze also the non-activated counterpart of all the subsets of CD4 and CD8 T cells. Does this mean that all TSCM are DR+ CD38+ and PD1+?

In figure legend2 (line 208) the title says CD8+ T cells, but the figure is about CD4 T cells.

Finally, in material and methods the authors should add more details about HIV-DNA quantification. They refer to other papers and there is a wide literature on the subject, but they should add at least the sequence of the primers.

Author Response

Reviewer 1

In this manuscript, Lo Tartaro and colleagues analyze T and B cell population 22 HIV patients during the acute phases of infection and follow them 2 and 12 months after initiation of therapy.

Determining how T and B cell populations changes after therapy is an important topic for the field. Moreover, the cohort, even though small, is very peculiar as not many studies are performed in patients during acute HIV infection.

The authors show how the most significant changes in CD4 T cell subsets composition occur 12 months after therapy while changes in CD8 are already present 2 months after treatment. Specifically, they observed a reduction of activated memory and exhausted activated TSCM in the CD4 T cell compartment, while the reduction in activated CD8 was detectable at the early time point. Significant changes were also observed for B cells 12 months after initiation of ART.

Of note, a correlation between viral load and a specific subset of cytotoxic CD4 effector memory cells as well percentage of IgM plasmablast and CD4 T cell count was observed.

This study is well done, and the data support the conclusions.

Comment 1: However, the manuscript is hard to follow due to the lack of information in the introduction. The authors should improve it. In paragraph 3.3.1_2_3, Lo Tartaro and colleagues describe all the subsets they found through clusterization, what are those? Why did they choose those PD-1, CD38 and CD57. A little bit of background would help the reader to understand the importance of their findings.

We thank the reviewer for the comment and we improved the introduction as requested.

Comment 2: Why did the authors completely disregard monocytes? Monocytes contribute to both HIV persistence and pathogenesis.

Monocytes and macrophages are significant mediators of inflammation, and dysregulation of their inflammatory functions during HIV infection is a key driver of comorbidities. Despite that, the limited number of cells per sample did not allow detailed investigation on the monocytes subpopulations and phenotype.

Comment 3: With the exception of TSCM, the authors analyze also the non-activated counterpart of all the subsets of CD4 and CD8 T cells. Does this mean that all TSCM are DR+ CD38+ and PD1+?

We thank the reviewer for this comment that allow us to explain this issue more in details. Unsupervised clustering applied to the data obtained by studying HIV+ individuals did not allow to reveal such small subpopulation of TSCM, that has an extremely low frequency. Thus, here we are able to show only the most represented subpopulation of TSCM, whose phenotype is mainly that of activated cells. However, by manual gating, we investigated TSCM phenotype. As displayed below, within TSCM cells (defined as viable, CD3+, CD4+,CD45RA+,CCR7+,CD27+,CD95+,CD28+), we identified two main phenotypes: 70% are HLA-DR+,CD38+,PD1+, while nearly 15% are not activated. We also checked this percentages in HIV- individual and found that most of them were CD38-,HLADR-, i.e., not activated. As you can see in the figure here below, the number of CD38+,HLADR+ cells was too small to allow any further consideration on PD1 expression.

Gated on CD3+CD4+CCR7+CD45RA+CD27+CD28+

HIV-

HIV+

Comment 4: In figure legend2 (line 227) the title says CD8+ T cells, but the figure is about CD4 T cells.

We apologize for the mistake. We amended as suggested.

Comment 5: Finally, in material and methods the authors should add more details about HIV-DNA quantification. They refer to other papers and there is a wide literature on the subject, but they should add at least the sequence of the primers.

As requested by the reviewer we added the missing information in material and methods (DNA extraction and quantification of HIV-DNA).

Reviewer 2 Report

In the study:

 Effective treatment of patients experiencing primary, acute HIV  infection decreases exhausted/activated CD4+ T cells and CD8+ 3T memory stem cells

the authors identify changes after 2 and 12 months of ART. In particular, among the rest:

 “T2, with a reduction of activated memory, cytolytic and activated/exhausted stem cell memory T (TSCM) cells. Changes were present among CD8+ T cells since T1, with a reduction of several activated subsets, including activated/exhausted TSCM. At T2 a reduction of plasmablasts and exhausted B cells was also observed. “ 

Some of the claims of this paper are already present in other studies, but it is valid to present an organic work, studying CD4, CD8, and B subpopulations at once, this was not done before.

From this point of few, the paper is valid and adds some new info to the field of HIV reservoir study and it deserves to be published.

Here are some minor corrections

Fig.1, 

In general, I don t like hist plots, it is always better to present the data as dot plots; In particular for fig1.c I would like to see a zoom-in of the viral load. In the clinical setting we consider 50 copies/mL as the max amount of virus to talk about successful viral suppression, In figure 1.c I cannot identify the copy numbers T2. This is an exclusion parameter, and any donor that is not suppressed below 50 copies/mL should be excluded from the study.

Fig2c and 3c.

Both of them should have a code color donor-specific. This is because many of your p values are driven by donor out layers. This is evident in fig2c EM38+DR+, EM38+DR+57+, TM 38-DR-, EM38+DR+, and so on. My point is, with the donor color-specific coding we can identify if those out layers are specific for one of a few donors or if they are randomly distributed between the donors. In the second case, this will reflect the normal inter donors' variability, but as of now, we cannot draw a conclusion.

Fig2D

It is clear that the populations here (red and green boxes) are part of the CD4+CCR7-CD45-CD27-CD28-PD1+CD57+ BUT the dot-plot is misleading. If I look at it, the populations in red should be around 80-90 % and not the 2-3%. You should rephrase this.

Gating strategy

In the supplementary data, you presented your gating strategy. I don t see here any mortality marker. The cells you used were not fresh isolated, so it is normal that after thawing you will have some dead/dying cells. You cannot exclude them just by gating strategy. In the material and method, you wrote about the use of LIVEDEAD staining, I would add it in the gating strategy figure (after all u add the time/promokine, so why not mortality???).

CD95

This is quite a tricky staining. 

Did you perform this staining at 4C???? CD95 induces apoptosis in target cells with consequent alteration of many other surface markers.

Did you use a healthy donor as CD95 control? Many pathologies alter the CD95 expression including HIV infection.

I would like to see a detailed CD95 staining protocol, as in PMID 23222456 , for example.

Bioinformatic

It is nice to see the use of some bioinformatic other than in sequencing studies.

The materials and method about it are well written but you forgot to add some details about the use, version, and so on, for UMAP and FlowSOM.

Author Response

Reviewer 2

The authors identify changes after 2 and 12 months of ART. In particular, among the rest:

 “T2, with a reduction of activated memory, cytolytic and activated/exhausted stem cell memory T (TSCM) cells. Changes were present among CD8+ T cells since T1, with a reduction of several activated subsets, including activated/exhausted TSCM. At T2 a reduction of plasmablasts and exhausted B cells was also observed. “

Some of the claims of this paper are already present in other studies, but it is valid to present an organic work, studying CD4, CD8, and B subpopulations at once, this was not done before.

From this point of few, the paper is valid and adds some new info to the field of HIV reservoir study and it deserves to be published.

Here are some minor corrections:

Comment 1: Fig.1, in general, I don t like hist plots, it is always better to present the data as dot plots; In particular for fig1.c I would like to see a zoom-in of the viral load. In the clinical setting we consider 50 copies/mL as the max amount of virus to talk about successful viral suppression, In figure 1.c I cannot identify the copy numbers T2. This is an exclusion parameter, and any donor that is not suppressed below 50 copies/mL should be excluded from the study.

As requested by the reviewer, we replaced the hist plots with dot plots. Furthermore, in figure 1c we added a zoom-in of viral load for T2 time point. Ticked line represent 50 copies/mL. All patients but one displayed a viral of 50 copies/mL. One patient displayed a viral load of 292 copies/mL. We have indicated this in the text and explained as a possible viral blip, likely not relevant from a clinical point of view since the patient did not show any other increase of the viral load during the following months.

Comment 2: Fig2c and 3c. Both of them should have a code color donor-specific. This is because many of your p values are driven by donor out layers. This is evident in fig2c EM38+DR+, EM38+DR+57+, TM 38-DR-, EM38+DR+, and so on. My point is, with the donor color-specific coding we can identify if those out layers are specific for one of a few donors or if they are randomly distributed between the donors. In the second case, this will reflect the normal inter donors' variability, but as of now, we cannot draw a conclusion.

We thank the reviewer for the comment, and we agree that it is important to determine whether the out layers are specific for one of a few donors or if they are randomly distributed. Thus, for this reason, we barcoded each sample with a specific color and observed that the out layers are randomly distributed - as reported here below.

Fig. 2c

Fig. 3c

Comment 3: Fig2D. It is clear that the populations here (red and green boxes) are part of the CD4+CCR7-CD45-CD27-CD28-PD1+CD57+ BUT the dot-plot is misleading. If I look at it, the populations in red should be around 80-90 % and not the 2-3%. You should rephrase this.

We apologize for the mistake. We corrected the percentages in the Fig 2D, as suggested.

Comment 4: (Gating strategy) In the supplementary data, you presented your gating strategy. I don t see here any mortality marker. The cells you used were not fresh isolated, so it is normal that after thawing you will have some dead/dying cells. You cannot exclude them just by gating strategy. In the material and method, you wrote about the use of LIVEDEAD staining, I would add it in the gating strategy figure (after all u add the time/promokine, so why not mortality???).

We used the LiveDead-RED for the cell sorting, while PromoFluor-840 (Promokine) was used for the immunophenotyping of T and B cells by flow cytometry. For this reason, we modified the Supplementary Figures 1 adding the gating strategy used to identify and sort viable naïve (CD45RA+CCR7+) and effector memory (CD45RA−CCR7−). Moreover, “Promokine” reported in the gating strategy of T and B cells has been replaced by “PromoFluor-840”.

Comment 5: (CD95). This is quite a tricky staining.

Did you perform this staining at 4C???? CD95 induces apoptosis in target cells with consequent alteration of many other surface markers.

Did you use a healthy donor as CD95 control? Many pathologies alter the CD95 expression including HIV infection.

I would like to see a detailed CD95 staining protocol, as in PMID 23222456, for example.

We added more detailed information about protocol staining in the materials and methods section. Note that in our protocol cells were never incubated at 4°C. In particular, the staining was performed in two steps: 1) Viability (PromoFluor-840) at room temperature for 20 minutes; 2) surface (Duraclone B or T plus drop-in antibodies) at room temperature for 20 minutes. Samples were acquired immediately. A staining buffer (50% of Brilliant Stain Buffer and 50% of FACS buffer) that contains sodium azide, was used to overcome CD95 induced apoptosis, as suggested in PMID 23222456.

Moreover, we used a healthy donor (HIV-) as CD95 control to set the gate and the correct cofactor for the unsupervised analysis, as reported below.

HIV-

Gated on CD3+CD4+CCR7+CD45RA+

Gated on CD3+CD4+CCR7-CD45RA-

HIV+

Comment 6: (Bioinformatic). It is nice to see the use of some bioinformatic other than in sequencing studies.

The materials and method about it are well written but you forgot to add some details about the use, version, and so on, for UMAP and FlowSOM.

We added as requested the version of UMAP (version 0.2.0) and FlowSOM (version 3.12). in the materials and methods section. Moreover, co-factors calculation has been added.
